# A pairwise randomised controlled trial of a peer-mediated play-based intervention to improve the social play skills of children with ADHD: Outcomes of the typically-developing playmates

Sarah Wilkes-Gillan[1], Reinie Cordier[2,3]*, Anita Bundy[4], Michelle Lincoln[5], Yu-Wei Chen[1], Lauren Parsons[3], Alycia Cantrill[1]

1 School of Health Sciences, Faculty of Medicine and Health, The University of Sydney, Sydney, New South Wales, Australia, 2 Department of Social Work, Education and Community Wellbeing, Faculty of Health and Life Sciences, University of Northumbria, Newcastle upon Tyne, United Kingdom, 3 Curtin School of Allied Health, Faculty of Health Sciences, Curtin University, Perth, Western Australia, Australia, 4 Department of Occupational Therapy, College of Health and Human Sciences, Colorado State University, Fort Collins, Colorado, United States of America, 5 Faculty of Health, University of Canberra, Canberra, Australian Capital Territory, Australia

* reinie.cordier@northumbria.ac.uk

**Data Availability Statement:** All relevant data are within the manuscript.

## Abstract

To examine the effectiveness of a play-based intervention for improving social play skills of typically-developing playmates of children with ADHD. Children (5–11 years) were randomised to an intervention ($n = 15$) or waitlisted control group ($n = 14$). The Test of Playfulness was scored by a blinded rater. Between-group statistics compared the change of the intervention (10-week intervention) and waitlisted control (10-week wait) groups. Change in the intervention group following intervention was significantly greater than the change in the waitlisted control group. When combining data from the groups, playmates' ($n = 29$) mean ToP scores improved significantly following intervention, with a large effect pre- to post-intervention and pre-intervention to follow-up. Typically-developing playmates of children with ADHD benefited from participation in a peer-mediated intervention.

## Introduction

The social difficulties experienced by children with ADHD are much greater than those of their typically-developing peers [1, 2]. Children with ADHD are often unpopular, frequently rejected by peers and have challenges in relationships with siblings and developing and maintaining friendships [3, 4]. For example, friends of children with ADHD have reported more conflict in their relationships with children with ADHD overtime, where children with ADHD have not reported the same deterioration in their friendship quality over the same 6-month period [5]. In the context of dyadic friendships, children with ADHD can report fewer friends and lower friendship stability and satisfaction than comparison children [1]. In an

**Funding:** We would like to acknowledge funding provided by the Rotary Club of Mosman and by the University of Sydney's Postgraduate Research Support Scheme. We would also like to thank the Australian Government for the provision of the Australian Postgraduate Award scholarship. The funders had no role in study design, data collection and analysis, decision to publish, or preparation of the manuscript.

**Competing interests:** The authors have declared that no competing interests exist.

observational study of dyadic play skills in children with ADHD, 60% of children with ADHD opted to participate with a sibling playmate because they could not identify a similar-aged peer who they considered to be a regular playmate [6]. Play is arguably the most important social context in which children initiate, develop and maintain friendships, and one explanation for the social relationship difficulties of children with ADHD is the observable differences in their social play skills.

Difficulties with empathy and perspective taking can explain the differences in the social play skills of children with ADHD [6], and can result in conflict, rejection, and poorer quality relationships with peers [3, 5]. When playing with friends, children with ADHD have been observed to break more rules during competitive play, appear more self-focused when negotiating, and lack perspective-taking skills [6, 7]. In contrast to control children, children with ADHD have also been found to violate rules in games more often, engage more frequently in self-centred, insensitive negotiations; all of which predicted deterioration in dyadic friendship quality over a 6-month period [5]. While some of this difficulty with friendships may be associated with the social skill challenges of children with ADHD, a proportion may also be associated with their social environments and opportunities to interact socially with peers. Teacher- and parent-report as well as observational data have demonstrated that compared to controls, children with ADHD often befriend peers who also have ADHD, oppositional symptoms and social skills deficits [6–8]. Some adults perceive parents have played a role in the manifestation of their child's ADHD-related behaviours through poor parenting (i.e., lack of discipline) and parents of children with ADHD feel shame and avoid social interactions for their child as a result [9–11]. Conversely, parents of typically-developing children may influence who their child plays with, not wanting their child playing with a child they perceive to be a negative influence, resulting in fewer opportunities for children with ADHD to interact socially with typically-developing peers.

In the absence of peer-friendships, siblings are often the most common playmate of children with ADHD [12]. Researchers studying the 'typically-developing' siblings of children with ADHD have observed difficulty with pro-social behaviour compared to control siblings, though not to levels that reach clinical significance, and siblings of children with ADHD can be at increased risk of emotional and behavioural disorders [6, 13, 14]. Nonetheless, while sibling relationships can be problematic for children with ADHD, such relationships provide opportunities to practice and develop social skills [6, 14].

The social challenges experienced by children with ADHD warrant the development of evidence-based interventions that focus on the social play skills of children with ADHD in conjunction with those of their usual playmates. Such interventions need to address the social, emotional and cognitive skills children need to successfully cooperate, negotiate, empathise and resolve conflict with peers so as to develop and maintain quality friendships. However, given that the regular playmates of children with ADHD may also face the same social challenges, albeit to a lesser extent, interventions that include those regular peers may be better placed to impact on children's social functioning, because they can impact on the social environment of children with ADHD rather than addressing the social difficulties of children with ADHD in isolation.

Peer inclusion interventions are an ideal way to address both the social play difficulties of children with ADHD and strengthen the play abilities of their regular playmates. The role of the peer within peer inclusion interventions can be categorised into three broad groups based on the way that peers are included: peer proximity, peer involvement and peer-mediation [15]. Proximity involves placing the target child within close distance to a purposefully selected, socially skilled peer with the assumption that interactions will naturally occur. In a peer involvement intervention, participants facilitate each other's learning. However, participants

often have similar skill levels and diagnoses [15]. Peer-mediation interventions are an extension of peer involvement; the peer is an active agent of change–using verbal prompts and gestures to encourage the target child to use the intervention strategies. Because peers naturally respond to the target child's behaviours, they require training to provide feedback when undesired behaviours occur [15, 16].

Two recent systematic reviews have evaluated peer inclusion interventions for children with ADHD. Across the reviews, peer inclusion was found to be advantageous for improving the social functioning of children with ADHD, compared to treatment as usual [15], and peer interactions within interventions were effective for improving play skills, communication (pragmatic language, joint participation) and social participation [16]. Peer interactions were also effective in reducing undesirable social behaviours (dominant behaviours, aggression) with improvements maintained over time in follow-up studies. While existing studies of peer inclusion for children with ADHD provide growing evidence of their benefits to children with ADHD, there is a distinct lack of literature evaluating peer outcomes following participation in peer involvement interventions [15].

The way in which peers have been included in interventions for children with ADHD has also varied across existing literature, with one review of 17 studies finding that 16 interventions used peer involvement, one used peer proximity and no study used peer mediation [15]. The lack of peer-mediated interventions (PMIs) for children with ADHD is somewhat surprising given that traditional social skills interventions continue to demonstrate limited effectiveness for this population of children [17–19]. Peer-mediation is an empirically supported approach to social skills intervention for children with autism [20], and researchers suggest PMIs may be particularly beneficial for children with ADHD who seek interaction as frequently as their peers but who have difficulty performing the necessary social skills as spontaneous peer interactions unfold [5–7, 21]. PMIs are based on the premise that peers have the capacity to motivate children and influence their behaviour [22], and peers have key roles in following instructions, implementing intervention strategies, providing feedback, modelling and reinforcing desired behaviours, providing an opportunity to practice target skills, and facilitating social interactions [20, 23, 24]. Moreover, interventions involving peers, and in particular PMIs, have the potential to benefit all peers involved due to the peer's active engagement in the intervention and the training they receive [16].

Since the aforementioned reviews of peer inclusion interventions for children with ADHD, a play-based PMI for children with ADHD was evaluated. The randomised controlled trial (RCT) demonstrated that the intervention was effective for improving the playfulness of children with ADHD, including play skills critical for social play [25]. Given that this play-based PMI achieved its primary aim of improving the playfulness of children with ADHD, further exploration of the intervention is warranted to understand the effects of the intervention from a more holistic perspective. Critical to the intervention was the inclusion of a typically-developing peer in every clinic-based intervention session as well as home-based activities between clinic visits. Throughout their participation peers were encouraged to support the play of children with ADHD, and model play behaviours that lead to mutual enjoyable social play experiences. Ensuring that there are also positive effects on the playfulness of these peers, or in the least there are no detrimental effects, is critical given that they and their families are dedicating resources to the intervention. Furthermore, should typically-developing playmates be recruited for this PMI in the future, an understanding of the profiles of the playmates who receive the largest benefit is critical to ensuring that children who are less likely to benefit are not recruited to an intervention that will not provide reward for the time spent participating. The aim of this current study was to examine the effect of the play-based PMI on the play skills of the peers involved, and to examine participant variables that predicted change.

The RCT protocol, outcomes of children with ADHD, and parents' treatment adherence are reported in detail by Wilkes-Gillan and colleagues [25], however, in the interest of completeness the intervention approach is described briefly here. Children with ADHD attended six clinic-based intervention sessions over a period of 10 weeks and invited a typically-developing peer to attend as a playmate. The sessions combined video self-modelling (in the form of video feedback and feedforward) with peer- and therapist-modelling in the context of child-led free play. Between session parents were provided with a manual to read and video to watch with their child. Parents of children with ADHD also arranged a playdate for their child and peer between clinic sessions. The primary objective of this study was to understand whether participating in this PMI had a positive effect on the play skills of playmates. Secondary objectives were to understand whether changes in playmate's play skills were maintained in the short term and generalised to a new setting, and the playmate behavioural traits that were associated with greatest change. Using the Test of Playfulness [26] we tested the following hypotheses:

1. Over a 10-week period, the change in overall play skills of playmates who attended the play-based PMI will be significantly greater than the change in overall play skills of playmates in a control group who have not attended the intervention;

2. The overall play skills of playmates will improve significantly from pre- to post- intervention, with improvements maintained one month later; and

3. Test of Playfulness items related to social play will improve significantly for playmates from pre- to post-intervention and generalise to the home environment.

4. Improvements in playmate's play skills over the intervention period will be associated with their play skills and behavioural profiles at baseline.

## Methods

Here we report the playmate outcomes from a two-group parallel trial that formed part of a larger study [25]. The Consolidated Standards of Reporting Trials (CONSORT) statement guided the reporting of this trial [27]. The trial protocol was registered with the Australian New Zealand Clinical Trials Registry and approved by the University of Sydney Human Research Ethics Committee (approval number: 2013/109). In this single site randomised controlled trial (RCT), participants were randomly assigned to an intervention or waitlisted control group. The intervention group received a 10-week play-based PMI and the control group received no treatment for 10-weeks, receiving play-based PMI thereafter.

### Participants

Following ethical approval participants were recruited for this RCT via convenience sampling. Over 11 months, parents of 45 children with ADHD contacted the first author. Of these, 31 met the inclusion criteria and two ceased their involvement after baseline assessment leaving a total sample of 29 (Fig 1). Each child with ADHD identified a typically-developing playmate to participate with them. More information about participants with ADHD in the study is reported by Wilkes-Gillan and colleagues [25].

**Typically-developing playmates.** Playmate participants were aged 5–11 years, and were typically-developing peers or siblings who had weekly interactions with the child with ADHD. We included playmates known to the children with ADHD to promote friendships and provide continuing opportunities for social interaction away from the intervention sessions. Playmate participants had scores on the Conners Comprehensive Behavior Rating Scales [CCBRS;

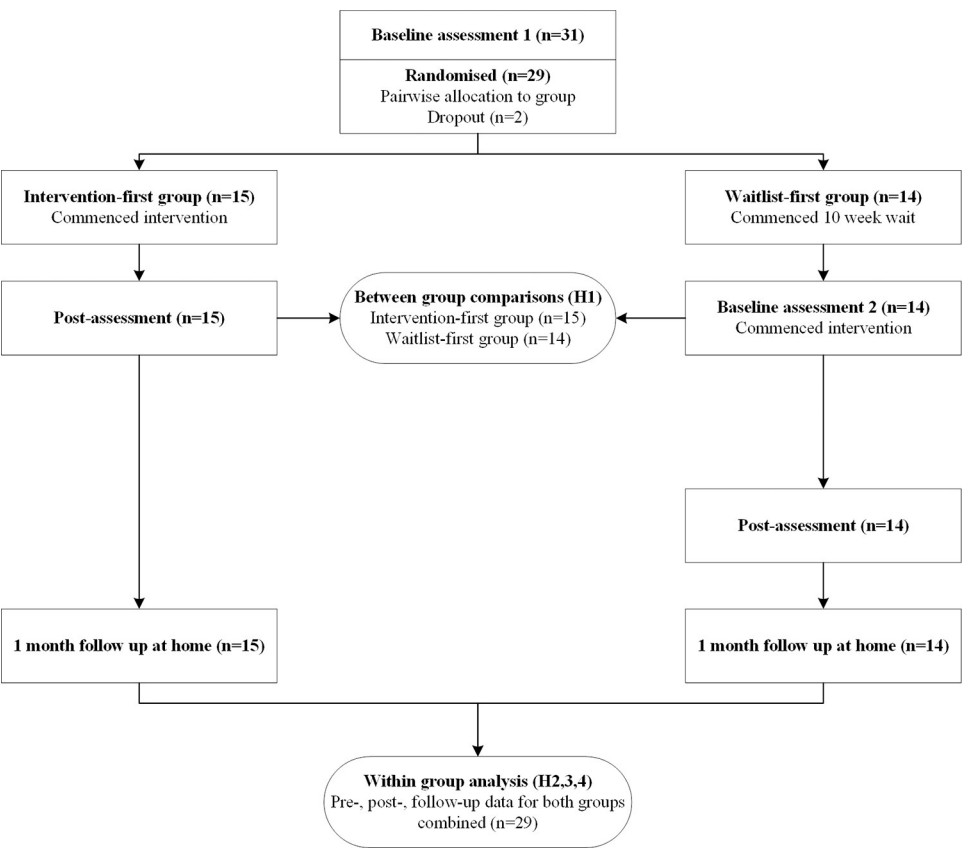

**Fig 1. CONSORT participant flow diagram.**

28] below the borderline clinical cut-off (i.e., T-scores ≤ 65 for DSM-IV subscales) indicating absence of ADHD symptoms or other diagnoses. They also obtained T-scores ≤ 65 for the behaviour, social and communication subscales. Parents reported that neither they nor their child's teacher had concerns about the playmate participant's social skills, behaviour or academic development. Parents and children over the age of 7 provided informed written consent for participation in the study. Children younger than 7 years provided verbal assent in the presence of their parents and researchers to account for the developing literacy skills of younger children.

### Instruments

**Test of Playfulness [ToP; 26].**   We used the ToP to examine playmates' play skills in peer-to-peer play interactions pre-, post-, and 1-month following the intervention. The ToP is a 29-item observation-based instrument scored on a 4-point scale to reflect extent, intensity, or skilfulness of play behaviours [26]. The ToP has evidence for excellent inter-rater reliability with data from 96% of raters fitting the expectations of the Rasch model; moderate test-retest reliability (intraclass correlation 0.67 at $p < .01$) and construct validity [data from 93% items and 98% of people fit Rasch expectations; 29]. The ToP can be used for children between 6 months and 18 years.

For this study, video footage of dyads (child with ADHD and their playmate) playing was recorded at three time points for the intervention group (baseline, post-, and 1-month following intervention), and four times for the waitlisted group (an additional baseline recording 10

weeks prior to intervention week 1). All footage was recorded in the playroom, except for the follow-up footage which was filmed at the home of the child with ADHD. An independent researcher who was blinded to study purpose and participant's group allocation viewed the footage to rate playmate's playfulness on al 29 ToP items. An overall, interval level ToP score for each child is obtained by converting raw, ordinal item ratings via Rasch analysis. Nine ToP items reflect social play skills that were of interest to this study: 1) *initiating* interactions, 2) *negotiating*, 3) *sharing*, 4) *supporting* a playmate, 5) time in *social* interactions, 6) intensity of involvement in *social* interactions, 7) skill of *interacting*, 8) *giving* verbal and non-verbal cues, and 9) *responding* to others' cues [25]. Playmate's quotes from conversations during play were also collected if they were relevant to the social play skills, as a qualitative demonstration of the expert ways that playmates used those skills to promote playful interactions with children with ADHD.

**Conners Comprehensive Behavior Rating Scales [CCBRS; 28].** The parent-rated CCBRS is a widely used screening tool for identifying symptoms consistent with diagnoses and behavioural difficulties in children. The CCBRS has excellent evidence for reliability and validity: Cronbach's alpha .67 to .97, test—retest reliability coefficient .56 to .96 ($p < .001$), and inter-rater reliability coefficients .50 to .89 ($p < .001$). The CCBRS has a mean classification accuracy of 78% across forms [28].

## Procedure

**Randomisation.** Participants were randomised to an intervention group, or waitlisted control group. As sporadic recruitment was expected, randomisation was conducted with a block size of two, with simple randomisation used to assign one of each two children who entered to each group (1:1 allocation ratio). Opaque envelopes containing slips of paper labelled 'group 1, intervention' or 'group 2, waitlist' were prepared and sealed by the first author. Once two parents had booked a baseline assessment, two sealed envelopes, one per group, were taken to an academic staff member not involved in the research. The person shuffled the envelopes and used a coin toss to allocate the sealed envelopes to the participants. Envelopes were opened to reveal group allocation at the conclusion of participants' baseline assessment [25]. reports the concealment and randomisation procedures for this RCT in full.

**Baseline assessment.** Researchers and participants were blinded to group allocation during the 1-hour baseline assessment that took place at a university research clinic. The baseline assessment involved each dyad playing for 20-minutes in a clinic playroom without an adult present. The play session was filmed using a wall-mounted video-camera while the therapist and parent observed from behind a one-way-mirror. Children were introduced to the space and the playroom rules: "have fun" and "come out if you need an adult." During the baseline assessment, the therapist closely observed the playmates' interactions to ensure their suitability for inclusion in the program. The playroom was consistently set up with the same variety of toys including: basket-ball, bowling, soft bat and ball games, cars, figurines, nerf guns, a tent, dress-ups, play-doh, a sand box, floor games (e.g., Snakes ad Ladders, Twister™) and toys from electronic games (e.g., Angry Birds™) [25].

**Intervention sessions at the clinic.** We held 1-hour sessions at the clinic in weeks 1–3, 5, 7 and 10. Each session involved a 20-minute video-feedback session followed by play in the playroom. During video-feedback the therapist showed dyads approximately 3 minutes of edited video footage of themselves playing together during their previous clinic session. Footage was coded for children as "green play" (i.e., footage of desired social interactions) and "red play" (i.e., interactions that required improvement). The relevant colour appeared on screen for children prior to viewing the footage, with accompanying text as feedback on actions pertinent to the footage (e.g., 'Great playing together' or 'We can play our friend's game too').

The therapist discussed the footage with the children using child-friendly terminology to assist them in identifying positive "green" actions that would make their play more fun (e.g., share ideas). Asking leading questions was a critical strategy to engage the cueing playmates in the conversation (e.g., 'What made that play lots of fun?', "You looked frustrated, what did you want your friend to do?"). The therapist then supported the children to identify three key actions to remember before entering the playroom (video-feed-forward); examples included "playing the same game," "telling your friend to stop," "listening to each other," "trying our friend's favourite game," and "sharing ideas when we play" [25].

During the week 2 and 3 play sessions the therapist supported the children to play cooperatively by modelling desired pro-social skills. These skills included sharing, perspective-taking, problem-solving, negotiating and responding to a playmate's verbal and non-verbal cues [25]. The therapist supported the children to negotiate when disagreements occurred and to reinforce key messages. For example, "What do you think we can change to make this 'green play' again?" The therapist assisted playmates to implement strategies in difficult play situations, "you're turning away—that play seems too rough! What can you tell your friend?" and to highlight consequences of actions in play to children with ADHD, "If you don't share any toys, I can't play".

In weeks 7 and 10, after children engaged in video-feedback, they played in the playroom without therapist support. This enabled evaluation of the playmates' abilities to implement strategies without therapist support.

**Intervention home-modules.** During weeks 4, 6, 8 and 9, parents of children with ADHD facilitated a 40-minute playdate at their home, inviting the typically-developing playmate to participate. Parents used three play cards: green (Great play! Keep going!), red (Let's stop and think), and purple (3 things to remember) and the feedback terminology used in the clinic session to give the children feedback before, during and after the playdate [25].

**Follow up.** One month after the intervention, the first author visited the homes of children with ADHD to video-record the dyads playing. The author spent 10 minutes talking with the children before a 20-minute play session was recorded [25].

**Data analysis.** Prior to conducting data analysis to assess our hypotheses we converted children's overall ToP raw scores into interval level scores at each time point using Rasch analysis in Winsteps [version 3.70.1; 30]. To conduct the analysis, raw data from this study was entered into an existing database that contained ToP scores of other children with ADHD and typically-developing children ($N$ = 406). Goodness-of-fit statistics for children and items were within the parameters set *a priori* ($MnSq$ < 1.4; standardised value $\leq$ 2).

The two discontinuing participants completed < 10% of the process and demographic data were incomplete, so these cases were excluded from the analysis. We entered interval level ToP scores and demographic data into SPSS [version 19; 31] for all further analyses.

*Between group comparisons at baseline*. Demographic data for the intervention and control groups were compared prior to testing the study's hypotheses. A Kolmogorov-Smirnov test indicated data were normally distributed, therefore paired samples *t*-tests were used to conduct between-groups comparisons of mean ToP scores and CCBRS data for the children with ADHD and playmates at baseline. We calculated Pearson Chi Squares to compare the difference of paired nominal demographic data (i.e., gender).

*Hypothesis 1*: *Difference in change between intervention and waitlisted control groups*. First, mean overall ToP scores for each group at entry to the study were compared using *t*-tests for independent samples to ensure no group difference were present at baseline. To compare changes in mean ToP scores over time a change score was calculated for participants using interval-level overall ToP scores. Baseline scores were deducted from post-intervention scores (intervention), and first baseline scores from second baseline (waitlisted control group). As

data were normally distributed, *t*-tests for independent samples were used to compare the change in overall play skills of the intervention group over the 10-week intervention period with the change in overall play skills of the waitlisted control group during their 10-week wait. Significance levels were set at $p < 0.05$. Additionally, a *t*-test for dependent samples was used to compare the change for the control group from baseline one and two over the 10-week wait period.

*Hypothesis 2*: *Overall changes in children's social play skills and maintenance.* As testing hypotheses 2–4 required within-groups analyses, the pre-, post- and follow-up ToP scores for all participants (*n* = 29) were combined to increase the statistical power for the remaining analyses. According to G*Power [version 3.1.9.2; 32], a sample of *n* = 30 was required ensure adequate power based on the following parameters: 1) desired power (0.8); 2) statistical test (ANOVA); 3) alpha value (0.05), and 4) expected effect ($> 0.5$ large). The expected effect was based on pilot studies of the intervention [33, 34].

A repeated-measures one-way ANOVA to compared changes in the playmates' overall interval level ToP scores across the three time points. Complete data were available for all 29 children and Mauchly's test indicated the assumption of sphericity had not been violated. Post hoc Fisher's Least Significance Difference (LSD) tests were used to compare playmates' play skills from pre- to post-intervention, post-intervention to 1-month follow up, and pre-intervention to 1-month follow up. Significance levels were set at $p < .05$ and Cohen-*d* effect sizes were calculated by: group (time point mean—time point mean)/pooled SD for group measure scores. Effect sizes were interpreted as: small $\geq .20$, medium $\geq .50$, or large $\geq .80$ [35].

*Hypothesis 3*: *Changes in social ToP items.* We calculated changes and effect sizes for the raw ordinal scores of the nine ToP items associated with social play from pre-, to post- and 1-month following intervention. As raw item scores are ordinal level data and not normally distributed, we used non-parametric tests for analyses of ToP social items. Friedman tests calculations examined changes in each social ToP item mean scores across all time points. Significance was set at $p < 0.05$.

The *r* effect size was then used to calculate the effect sizes for non-parametric social ToP item data. The effect size (i.e., *r*), is obtained by dividing the Wilcoxon *Z* score by the square root of the sample size (i.e., 29); $r = Z / \sqrt{N}$ [32]. Cohen's guidelines for *r* are: small effect $\geq .1$, medium effect $\geq .3$ or large effect $\geq .5$ [32, 35]. To obtain the Wilcoxon signed rank tests for related samples, ToP social item scores were compared pre- to post-, post- to follow up, and pre- to follow up. We applied a Bonferroni correction to control the false discovery rate associated with multiple statistical tests. Applying this correction, we set a new familywise significance threshold by dividing the overall 0.05 significance level by the number of Wilcoxon tests performed within each time group comparison [i.e., 9; 36].

*Hypothesis 4*: *Playmate variables associated with intervention change.* We calculated Pearson's correlation coefficients to identify child-related variables associated with playmates' pre- to post-intervention change scores. The child-related variables included: pre-test ToP scores and *T*-Scores of the CCBRS scales.

## Results

### Between group comparisons at baseline

There were no statistically significant differences between intervention and waitlisted control playmate on parent demographic and ADHD symptomology. Child demographic variables were also comparable with the exception of gender; there were more male playmates in the intervention group. A majority of playmates were siblings of the children with ADHD (55%), with the rest of the sample comprised of cousins (7%) and friends (38%). On average, the age

**Table 1. Participant demographics.**

| | Intervention Group | | Control Group | |
|---|---|---|---|---|
| Parent Demographic Variables[a] | *ADHD* | *Playmate* | *ADHD* | *Playmate* |
| Mean age in years (SD) | 41.7 (7.0) | 42.0 (4.0) | 41.5 (6.0) | 43.0 (4.2) |
| Born in Australia | 8 of 12 | 8 of 12 | 10 of 13 | 8 of 13 |
| Qualifications: degree or diploma | 93% | 93% | 87% | 100% |
| Occupation: requires tertiary qualifications | 60% | 47% | 57% | 64% |
| Child Demographic Variables | *ADHD* | *Playmate* | *ADHD* | *Playmate* |
| Mean age in years and months (SD) | 8.2 (1.5) | 8.5 (1.9) | 8.5 (1.7) | 7.9 (2.3) |
| Male | 13 of 15 | 10 of 15* | 12 of 14 | 3 of 14* |
| Born in Australia | 14 of 15 | 14 of 15 | 12 of 14 | 13 of 14 |
| ADHD Symptomology (CCBRS)[b] | | | | |
| Hyperactivity symptoms | 75[c] (13.0) | 49 (11.0) | 74[c] (12.8) | 50 (7.9) |
| Inattention symptoms | 80[c] (11.7) | 53 (10.8) | 81[c] (9.8) | 50 (9.4) |
| Oppositional behaviour | 75[c] (13.4) | 59 (14.6) | 76[c] (13.0) | 52 (11.0) |
| Generalized anxiety disorder | 71[c] (11.5) | 54 (7.8) | 73[c] (12.9) | 51 (9.9) |
| Social problems | 75[c] (15.0) | 50 (6.7) | 81[c] (13.7) | 51 (11.2) |
| Language problems | 64 (14.2) | 46 (7.5) | 63 (10.5) | 50 (11.3) |
| Reason for playmate selection | | | | |
| Friend–Similar interests to target child | - | 5 of 15 | - | 6 of 14 |
| Sibling–Regular availability | - | 8 of 15 | - | 8 of 14 |
| Cousin–No siblings or friends identified | - | 2 of 15 | - | 0 of 14 |
| Age difference in child dyad, years/months | - | 1.8 (1.2) | - | 1.9 (1.5) |

[a] Some mothers enrolled more than one child in the program. Demographic information is therefore reported on 25 mothers of children with ADHD and 26 mothers of playmates.

[b] The CCBRS was used to confirm the diagnosis of ADHD.

[c] Mean scores were above the clinical cut-off, T-scores $\geq$ 70 on the DSM-IV subscales for children with ADHD. Playmates scored below the borderline clinical cut-off (T-scores $\leq$ 65) on all subscales.

* Significant difference was found between the ADHD (intervention vs. control) and playmate (intervention vs. control) groups across all interval level (i.e., CCRBS scores; *t*-tests), and nominal data variables (i.e., gender; Pearson Chi Square). There were significantly more male playmates in the intervention group ($p = .04$).

difference between playmates and children with ADHD was less than two years. Demographic information is reported in Table 1.

## Hypothesis 1: Difference in change between intervention and waitlist groups

The overall play skills of playmates in the intervention group and the waitlisted control group were not significantly different at baseline ($t = -1.727$; $p = .108$). The mean baseline score of playmates in the intervention group was 47.0 ($sd = 9.7$; range = 64.4–32.6). The mean baseline score of playmates in the control group was 54.7 ($sd = 11.9$; range = 69.4–32.4).

The change in the overall play performance of the playmates in the intervention group during their intervention phase (pre- to post-intervention) was significantly greater than the change in the overall play of playmates in the waitlisted control group during their 10-week wait period ($t = 5.93$, $p < .001$). The mean change in overall ToP scores for the intervention group was 24.9 ($sd = 9.6$; range = 40.2–5.4). The mean change in the overall ToP scores for the intervention group was -6.4 ($sd = 14.6$; *range* = -35.7–5.8).

For the intervention group, there were no significant differences in the playmates' social play skills over the 10-week period of no intervention ($t = -.1.67$, $p = .117$). The mean of the

first baseline score was 54.7 ($sd$ = 11.9; range = 73.7–32.4). The mean of the second baseline score was 48.2 ($sd$ = 11.3; range = 68.7–27.8).

## Hypothesis 2: Overall changes in playmate's play outcomes and maintenance

There was a significant main effect of time on the overall ToP measure scores for the playmates following the intervention, $F(2, 27)$ = 66.5 ($p$ < .001). Post hoc LSD analysis indicated playmates' overall play scores improved significantly from pre- to post-intervention with a large effect size detected: mean pre- = 47.3 ($sd$ = 10.3), mean post- = 69.2 ($sd$ = 8.6; $p$ < .001, $d$ = 1.5). There also was a large and significant difference from pre-intervention to the 1-month follow up: mean pre- = 47.3 ($sd$ = 10.3), mean follow-up = 69.0 ($sd$ = 6.1; $p$ < .001, $d$ = 1.6), indicating intervention effects were maintained for at least one month. We found no difference from post-intervention to the 1-month follow-up: mean post = 69.2 ($sd$ = 8.6), mean follow-up = 69.0 ($sd$-6.1; $p$ = 1.00, $d$ = 0.0).

## Hypothesis 3: Changes in playmates' social ToP item scores

There was a significant main effect of time for all nine ToP item scores relating to social play across the three points of measurement. Post hoc analysis indicated ToP social item scores improved significantly from pre- to post-intervention and from pre-intervention to the one-month follow up. No difference was found from post-intervention to the one-month follow up (see Table 2).

**Table 2. Playmates' changes in ToP social skill item scores over time.**

| | | Descriptive Statistics | | | | | | Friedman's[c] | | Post Hoc Pairwise Comparison[d] | | | |
|---|---|---|---|---|---|---|---|---|---|---|---|---|---|
| | | Pre | | Post | | Follow up | | Pre-post-follow up | | Pre to post | | Pre to follow up | |
| ToP Item[a] | Brief Item Description | Med | IQR[b] | Med | IQR | Med | IQR | $\chi^2$ | $p$ | $\chi^2$ | $p$ | $\chi^2$ | $p$ |
| Initiates | The child's skill/ability to initiate a new activity with another | 1.0 | 2.0 | 3.0 | 1.0 | 2.0 | 2.0 | 16.689 | < .001 | .897 | .002 | .707 | .021 |
| Negotiates | The child's skill/ability to negotiate with others using 'give and take' | 1.0 | 2.0 | 3.0 | 1.0 | 3.0 | 1.0 | 35.299 | < .001 | 1.190 | < .001 | .983 | < .001 |
| Shares | The child's skill/ability to allow others to use toys or ideas about the game | 1.0 | 2.0 | 3.0 | 0.0 | 3.0 | 0.0 | 36.500 | < .001 | 1.121 | < .001 | 1.103 | < .001 |
| Supports | The child's skill of helping others; using verbal support or by physical assistance | 1.0 | 2.0 | 3.0 | 1.0 | 3.0 | 1.0 | 43.471 | < .001 | 1.224 | < .001 | 1.362 | < .001 |
| Social extent | The extent/proportion of time the child interacts with others | 2.0 | 2.0 | 3.0 | 0.0 | 3.0 | 0.0 | 33.969 | < .001 | .948 | < .001 | 1.017 | < .001 |
| Social intensity | The intensity/depth of the child's interactions with other's during play | 1.0 | 1.0 | 3.0 | 1.0 | 3.0 | 0.0 | 43.687 | < .001 | 1.207 | < .001 | 1.328 | < .001 |
| Social skill | The child's skill/ability to interact with others in cooperative and competitive play | 1.0 | 2.0 | 3.0 | 1.0 | 3.0 | 1.0 | 42.333 | < .001 | 1.310 | < .001 | 1.379 | < .001 |
| Gives cues | The child's skill/ability to give verbal and non-verbal cues to others | 3.0 | 0.0 | 2.0 | 1.0 | 3.0 | 0.0 | 26.778 | < .001 | .776 | .009 | .828 | .005 |
| Responds to cues | The child's skill/ability to respond to others' verbal and non-verbal cues | 2.0 | 1.0 | 3.0 | 1.0 | 3.0 | 0.0 | 31.649 | < .001 | .914 | .002 | 1.103 | < .001 |

[a] Items can be rated on skill, extent and intensity (degree).

[b] IQR = Interquartile range.

[c] Friedman's two-way ANOVA.

[d] Post hoc pairwise comparison tests $p$ = adjusted $p$-value after post hoc Dunn-Bonferroni test. No post to follow up differences were statistically significant.

The changes in these social play items were demonstrated through children's conversations. Playmates were better able to negotiate to get their own needs met in play, such as expressing a sense of inequality and offering ways to overcome that (e.g., "it's not fair if we always play your game. I get to choose one too this time"). They were also able to demonstrate strategies to support the play of children with ADHD, reminding them of ways to promote mutually enjoyable play (e.g., "It's more fun sharing ideas about our game, isn't it", "That's a tricky rule–I didn't think of that one"). Playmates were also able to give clear cues to children with ADHD to indicate enjoyment or disapproval of the play experience in the moment ("That's too rough, stop or I'm not playing!").

### Hypothesis 4: Variables that correlated with intervention change

Four child-related variables were moderately correlated with playmates' pre- to post-intervention change scores. Playmates' baseline ToP score had the strongest correlation with pre- to post-intervention change ($r = -.690$, $p < .001$). Fig 2 shows the relationship between baseline ToP scores and ToP change scores and indicates that lower ToP scores at baseline were associated with larger change scores. Three variables from the CCBRS has a moderate negative correlation with playmates' change scores: $T$-Scores on the scales of Generalised Anxiety Disorder ($r = -.500$, $p$ .006), Social Problems ($r = -.457$, $p$ .013) and Major Depressive Episode ($r = -.398$, $p$ .032). That is, higher $T$-Scores (i.e., displaying more symptoms) on the above scales were correlated with smaller changes in ToP scores for playmates. Correlations for the remaining CCBRS scales were not significant.

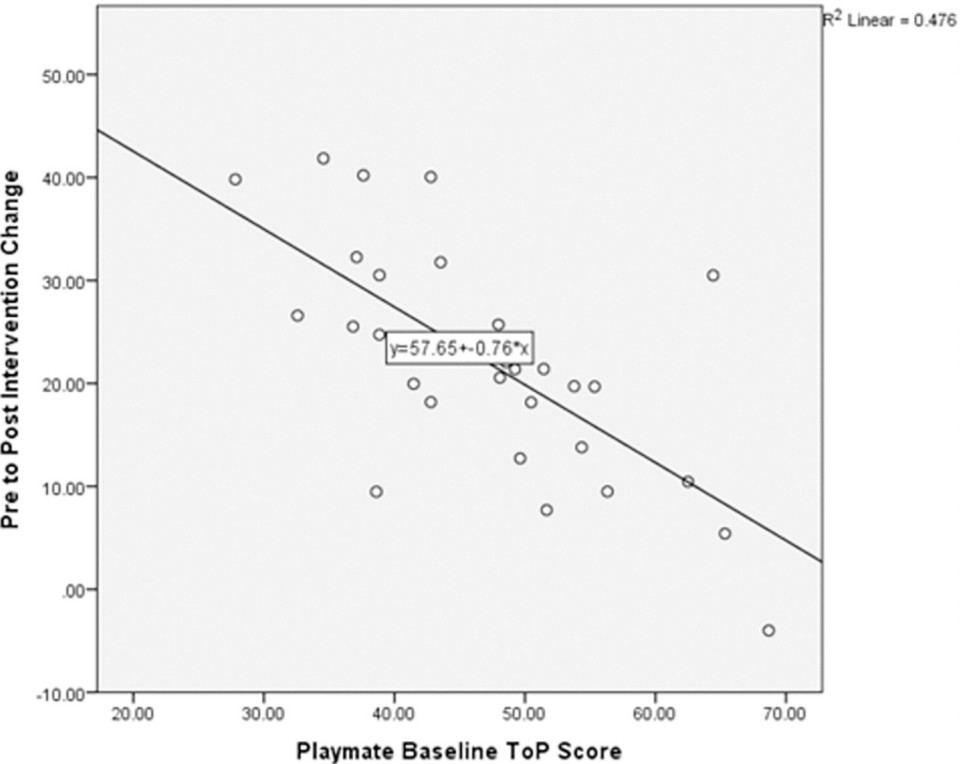

**Fig 2. Variables that correlated with intervention change.**

## Discussion

We investigated the play outcomes of typically-developing children involved in a play-based PMI for children with ADHD. Our findings showed that the intervention had a large, positive effect on the social play skills of typically-developing playmates of children with ADHD, and that effect was maintained 1-month later. Given that the intervention effect on playfulness from this play-based PMI is not limited to children with ADHD, these benefits to playmates strengthen calls in the literature for the inclusion of children's usual playmates in PMIs for children with ADHD [15, 16]. Improvements in playmate's play skills are critical for the social outcomes of these playmates and their peers with ADHD alike. As a result of these improvements, the preferred playmates of children with ADHD are likely to be more adept at using play-based strategies to counteract the challenging play behaviours often presented by children with ADHD, and maintain ongoing and mutually enjoyable social interactions [25].

Our results showed that after participating in the intervention typically-developing playmates had an improved ability in initiating play, negotiating and sharing, cooperating, giving and responding to social cues, supporting the play of their peer with ADHD, and maintaining a play interaction for longer and with greater intensity. Playmates were better able to negotiate to get their own needs met in play and successfully navigate challenging play situations, such as rough play. Wilkes-Gillan and colleagues [25] suggest that developing the social skills of the usual playmates of children with ADHD is likely to equip playmates to better engage in pro-social interactions with children with ADHD. For playmates, the changes in the particular social play skills measured within this study would likely offset the social challenges they are often faced with when playing with children with ADHD, and act as protective factors for their friendships with children with ADHD. While this study did not measure the dimensions of friendship for dyads at any stage, the associations between children's social play skills and friendship quality and duration would be an important area of future investigation.

Another critical finding of this study is that lower play scores before the intervention were associated with larger changes in play scores over the intervention period for playmates. This finding aligns with emerging evidence for PMIs that suggests peers who display low levels of social skills should not be included in PMIs [6, 8, 24, 37, 38]. In addition, smaller changes in play sores were associated with playmates who had higher behaviours symptoms of anxiety, depression, and social problem scores prior to the intervention. Commonly used inclusion criteria for peers in other PMI studies have included typical social and language development, absence of behaviour difficulties, an interest in interacting with the target child, and regular availability [20, 39, 40]. These criteria are often used to ensure benefits for the target child are maximised as playmates with such a profile are more likely to engage in and implement the strategies of the PMI. Given that in this study the playmates with fewer social problems and behavioural symptoms of mental health problems benefitted the most, these inclusion criteria also appear to be critical from the perspective of the playmates. We were unable to conduct more sophisticated analyses to explore the traits of the playmates in more detail due to sample size limitations. Research that identifies the traits of peers associated with optimal intervention outcomes for both children with ADHD and the peers themselves would progress our understanding of who the 'ideal' peers are to include in PMIs into the future.

As in this current study, Mikaimi and colleagues found the social benefits of a teacher-delivered intervention to promote a socially inclusive classroom extended beyond children with ADHD to the 113 typically-developing peers involved [41]. The benefits for the typically-developing peers in Mikaimi's study included reduced negative sociometric nominations from peers in the program, increased reciprocated friendships, and reduced negative interactions [41]. Further, positive outcomes were enhanced for typically-developing peers who had higher

levels of disruptive behaviour [42]. Similarly, we found that the playmates who benefited most from the intervention were more likely to have lower play scores at the outset of the intervention. This last finding may be particularly important for the preferred playmates of children with ADHD who have been found to exhibit more social challenges than other typically-developing children [5–8].

Increasing playmate's skills and ability to play with children with ADHD may have further downstream effects on the stigma experienced by children with ADHD and their families. If typically-developing playmates are better able to promote positive social interactions, this in turn may reduce their parent's worry or perceptions about them being negatively influenced by children with ADHD [9–11]. While playmates' and their parents' perceptions were not examined in this study, their perceptions and experiences following PMI is an important area for future research.

Continued research on PMIs is required to determine which peers are best suited for inclusion in such interventions [24, 43]. As in our study, PMIs for children with ASD have typically involved socially competent, typically-developing peers [39, 44]. However, results from our study showed that while clinically-speaking all playmates were typically-developing, the benefit to playmates was reduced for those with higher behavioural symptom scores for social problems, anxiety, or depressive episodes. In many previous studies involving children with ADHD, a large proportion of the peers had a diagnosis of ADHD [15]. Failure to include typically-developing peers in the interventions may have reduced the benefits for children with ADHD and their usual playmates alike.

## Limitations and future directions for research

Findings from the RCT support the use of and benefits to typically-developing children in PMIs for children with ADHD. However, the intervention was limited to dyadic interactions, and benefits to playmates from their own and their parents' perspectives were not explored. Further research is needed to determine if treatment effects generalise to other peers and social contexts such as school. It is also unknown if there are 'ideal' playmates and, if so, what characteristics are associated with being an 'ideal' playmate. However, from an ecological validity perspective, using playmates with whom the child with ADHD interacts with on a day-to-day perspective makes intuitive sense. The RCT had a small sample size which limited our ability to explore a more sophisticated moderation analysis to determine factors that influenced intervention outcomes, such as interactions between the play skills of the children with ADHD and the play skills and outcomes of their playmates. There is also a risk of selection bias due to the recruitment approach and inclusions criteria. Future studies should use larger sample size to unpack intervention moderators, and be implemented though services in the community so that the sample more closely reflects the broader population of children with ADHD.

## Conclusion

Outcomes for typically-developing peers should be investigated following participation in PMIs to ensure peers and target children alike benefit from participation. Findings from this RCT support the inclusion of typically-developing children in this play-based PMI, as participation has a positive effect on children's social play skills that is maintained in the short term after the intervention. The typically-developing children who benefitted most from participation in this PMI tended to have fewer behavioural symptoms of social problems, anxiety and depression, indicating that consideration of the behavioural profiles of peers is important when considering who to include in PMIs. Factors associated with the benefits to typically-developing peers following PMIs are still emerging. Future research is needed to further

investigate the characteristics of 'ideal' playmates for inclusion this play-based PMI, and the perceived benefits of participation from the perspectives of parents and typically developing peers themselves.

## Acknowledgments

We wish to extend our sincere gratitude to the participating families and the organisations that provided financial contributions. We also wish to thank Dr. Richard Parsons for his assistance with the statistical analyses and occupational therapy students, Charishma Kaliyanda and Natasha Cuffe for their assistance in the clinic.

## Author Contributions

**Conceptualization:** Sarah Wilkes-Gillan, Anita Bundy, Michelle Lincoln.

**Data curation:** Sarah Wilkes-Gillan, Alycia Cantrill.

**Formal analysis:** Sarah Wilkes-Gillan, Yu-Wei Chen.

**Funding acquisition:** Sarah Wilkes-Gillan.

**Methodology:** Sarah Wilkes-Gillan, Reinie Cordier, Anita Bundy, Michelle Lincoln, Yu-Wei Chen, Lauren Parsons.

**Project administration:** Sarah Wilkes-Gillan.

**Supervision:** Reinie Cordier, Anita Bundy, Michelle Lincoln.

**Writing – original draft:** Sarah Wilkes-Gillan.

**Writing – review & editing:** Sarah Wilkes-Gillan, Reinie Cordier, Anita Bundy, Michelle Lincoln, Yu-Wei Chen, Lauren Parsons, Alycia Cantrill.

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
