## [Decision Letter · Decision Letter 0]

6 Jul 2022

PONE-D-22-07013A Pairwise Randomised Controlled Trial of a Peer-Mediated Play-Based Intervention to Improve the Social Play Skills of Children with ADHD: Outcomes of the typically-developing playmates

PLOS ONE

Dear Dr. Cordier,

Thank you for submitting your manuscript to PLOS ONE. After careful consideration, we feel that it has merit but does not fully meet PLOS ONE’s publication criteria as it currently stands. Therefore, we invite you to submit a revised version of the manuscript that addresses the points raised during the review process.

We look forward to receiving your revised manuscript.

Kind regards,

Amanda A. Webster

Academic Editor

PLOS ONE

https://journals.plos.org/plosone/s/file?id=ba62/PLOSOne_formatting_sample_title_authors_affiliations.pdf".

Reviewers' comments:

Reviewer's Responses to Questions

**Comments to the Author**

1. Is the manuscript technically sound, and do the data support the conclusions?

Reviewer #1: Yes

Reviewer #2: Yes

2. Has the statistical analysis been performed appropriately and rigorously? 

Reviewer #1: Yes

Reviewer #2: Yes

3. Have the authors made all data underlying the findings in their manuscript fully available?

Reviewer #1: No

Reviewer #2: Yes

4. Is the manuscript presented in an intelligible fashion and written in standard English?

Reviewer #1: No

Reviewer #2: Yes

5. Review Comments to the Author

Reviewer #1: This is an interesting study. looking at peer-mediated interventions in ADHD children.

A few comments, worth address and clarifying.

A general comment would be to make it clear in the manuscript, the primary and secondary objectives.

It was a bit challenging to navigate through the design, in that at the end the whole sample was combined and within groups comparisons were compared. This would suggest a cross-over design was implemented, it was hard to understand the randomisation arms, i.e. intervention first, control - first. Perharps to help reader it might be worth expanding on this more.

1. The authors have stated that randomisation was conducted with block size of two and using simple randomisation. Also the authors state a person shuffled the envelopes. More details should be provided, was there a randomisation schedule created, and if it was with block size and concealment via opaque envelopes why was there a need to “shuffle” the envelopes. A risk of selection bias here is possible and worth considering.

2. The sample size calculation has incomplete information to make it reproducible. I.e based on an effect size>0.5, is vague might need to put the exact. Also missing information about the variability.

3. Demographic information Table1 to appear in the results section.

Reviewer #2: The study reported in this manuscript is a secondary sub-study of a previous intervention study in children with ADHD. This sub-study is aiming to examine the play skills of the peer mediators in a previously reported peer mediated intervention study for children with ADHD. In some respects, this is looking at the fidelity of the approach (ie, the fidelity of the peer mediators peer mediators). I do wonder why this wasn't addressed as part of the report of the publication from the initial seed study?

Introduction: There are a number of statements within the introduction that need supporting references. page 3, line 52; page 3 line 54; page 3, line 73. page 5, line 109

Line 79 " As a reader I'm not convinced by the afore introduction that social challenges warrant the development of interventions to increase play participation. "

As a reader I'm not convinced by the afore introduction that social challenges warrant the development of interventions to increase play participation. So far the argument in the introduction has stated difficulties in social situations and friendships but there is no introduction about children with ADHD NOT participation in play - just that they have difficulties with social interactions during play, or may choose, or be directed by families, to play within their own family social situations. There is some evidence that solitary play is predictive of reduced anxiety and depression in ADHD, and also some evidence that children with adhd are more fun. Can the authors include a balanced view in their introduction, and particularly identify that it is children with ADHD AND difficulties with making and retaining friendships that was the focus of their research.

METHODS: Well described. Page 8, line 180-181 "Neither parents nor teachers raised concerns about the playmate

participant’s development.

Were parents and teachers specifically asked about behaviours of the playmate or are you saying that they just didn't rise any concern. There is a difference and this should be made clear. For example, parents and teachers may not raise concerns voluntarily because it would be inappropriate to share such information.

Instrument Test of Playfulness: Is there a value for a clinically meaningful change score since this is your main outcome of interest? Even more useful would be a value of worthwhile change. Is the change been anchored against a worthwhile change outcome.

Procedure and Intervention well described.

RESULTS: As per above: Is there a clinically meaningful change for ToP? This would be helpful in interpreting the change scores from a clinical perspective. Similarly it would be good to understand a worthwhile change? Was the ToP anchored to any assessment for this? I am not clear on teh value of looking at the playmates change scores r correlation with their CCBRS since their CCBRS is not within the clinical range, and the correlations are not strong (not unexpected since they are not within the clinical range.

DISCUSSION: The discussion focusses too much on the ADHD population rather than the main am of this sub study.

Line 413 -414 is is one of the first studies to report on the intervention outcomes of the peers of children with ADHD following a PMI."

Just because something is do e first doesn't mean it is of value. It doesn't add any value to your study by stating this so I would remove it. How does improvements in play of typically developing children support including them in interventions for ADHD? please elaborate on your statement around this.

Line 421-423 " As a result of these improvement, the preferred playmates of children with ADHD are likely to be better equipped to deal with the challenging behaviours often presented by children with ADHD to maintain ongoing and mutually enjoyable social interactions"

I don't agree with this statement. There is no outcome showing that these playmates become better equipped to deal with challenging behaviour since this was not measured in this study. All that you can say is that the intervention is effective in typically developing children

Line 426 -428 The benefits for the typically developing peers included reduced negative sociometric nominations from peers in the program, increased reciprocated friendships, and reduced negative interactions.

Are these the results from your study? Or Mikami. It is not clear. If from Mikami how does it relate to your outcomes since you didn't report on these?

Line 435-483 These paragraphs should go first in your discussion since it was the main aim of this study. I suggest you move them all up. Remove the "quotations" from discussion. These are qualitative results that you have not addressed in methods or results. The last paragraph 484-490 This last paragraph in the discussion is not related to playmates outcomes so I can't see the relevance of including it.

Conclusion: The last lines 507-511 are not a conclusion of this study.

6. PLOS authors have the option to publish the peer review history of their article (what does this mean?). If published, this will include your full peer review and any attached files.

Reviewer #1: No

Reviewer #2: No

---

## [Author Response · Author response to Decision Letter 0]

11 Aug 2022

Comment 1. Please ensure that your manuscript meets PLOS ONE's style requirements, including those for file naming. The PLOS ONE style templates can be found at

https://journals.plos.org/plosone/s/file?id=ba62/PLOSOne_formatting_sample_title_authors_affiliations.pdf".

Response: We believe we have made all of the appropriate style adjustments based on the templates provided.

Comment 2. Please include your full ethics statement in the ‘Methods’ section of your manuscript file. In your statement, please include the full name of the IRB or ethics committee who approved or waived your study, as well as whether or not you obtained informed written or verbal consent. If consent was waived for your study, please include this information in your statement as well.

Response: The name of the IRB is now detailed within the manuscript (Line 173)

Details for obtaining consent are provided on page 8 (Line 197-200)

Comment 3. Please review your reference list to ensure that it is complete and correct. If you have cited papers that have been retracted, please include the rationale for doing so in the manuscript text, or remove these references and replace them with relevant current references. Any changes to the reference list should be mentioned in the rebuttal letter that accompanies your revised manuscript. If you need to cite a retracted article, indicate the article’s retracted status in the References list and also include a citation and full reference for the retraction notice.

Response: Retracted citations referred to the seminal study that this paper is following up upon. We had removed them to maintain double blinding in the peer review process. We now understand that PLOS One has a single blind review process and these citations and references are now presented within the manuscript. 

Reviewer #1: This is an interesting study. looking at peer-mediated interventions in ADHD children. A few comments, worth address and clarifying.

1. A general comment would be to make it clear in the manuscript, the primary and secondary objectives.

Response: The following text has been inserted at the end of the introduction to articulate the objectives of the study (Lines 153-157).

The primary objective of this study was to understand whether participating in this PMI had a positive effect on the play skills of playmates. Secondary objectives were to understand whether changes in playmate’s play skills were maintained in the short term and generalised to a new setting, and the playmate behavioural traits that were associated with greatest change.

2. It was a bit challenging to navigate through the design, in that at the end the whole sample was combined and within groups comparisons were compared. This would suggest a cross-over design was implemented, it was hard to understand the randomisation arms, i.e. intervention first, control - first. Perharps to help reader it might be worth expanding on this more.

Response: The main objective was tested by randomising participants to an intervention or control arm of the study. Outcomes for the two groups were compared to address hypothesis 1.

Because the control group eventually went on to also receive the intervention after their waitlisted period ended we were able to pool the pre-, post, and follow up data from all participants to increase the power of the between groups analysis to address hypotheses 2-4. 

The CONSORT diagram (fig. 1) has been simplified to demonstrate the two arms of the study more clearly. The justification for pooling the pre, post and follow up data is provided within the Hypothesis 2 data analysis description (Line 328-331).

3. The authors have stated that randomisation was conducted with block size of two and using simple randomisation. Also the authors state a person shuffled the envelopes. More details should be provided, was there a randomisation schedule created, and if it was with block size and concealment via opaque envelopes why was there a need to “shuffle” the envelopes. A risk of selection bias here is possible and worth considering.

Response: The randomisation procedure has been revised (Lines 234-244).

We saw it important to shuffle envelopes prior to allocating them to a participant to ensure that envelopes containing group numbers were not always in the same order prior to the coin toss allocation procedure. 

We agree these is a risk of selection bias within this study, due to the recruitment approach and inclusion criteria rather than randomisation procedure. This is now acknowledged within the limitations (Line 529)

4. The sample size calculation has incomplete information to make it reproducible. I.e based on an effect size>0.5, is vague might need to put the exact. Also missing information about the variability.

Response: The exact effect size entered for the sample size calculations was 0.5; this has been amended in the manuscript (Line 333).

G*Power does not require parameters around variability to calculate the sample size required to detect a particular effect.

5. Demographic information Table1 to appear in the results section.

Response: Table 1 and results of the between-group comparisons for demographic variables are now reported under Results.

Reviewer #2:

1. The study reported in this manuscript is a secondary sub-study of a previous intervention study in children with ADHD. This sub-study is aiming to examine the play skills of the peer mediators in a previously reported peer mediated intervention study for children with ADHD. In some respects, this is looking at the fidelity of the approach (ie, the fidelity of the peer mediators peer mediators). I do wonder why this wasn't addressed as part of the report of the publication from the initial seed study?

Response: The primary aims of the initial study were to establish that the intervention approach was feasible, appropriate and beneficial for the children with ADHD. Given that initial findings indicated that this was the case, we determined that a more holistic evaluation of the intervention is warranted to ensure that there were also benefits (or in the least, no harm) to the playmates who were giving up their time to participate in the intervention as well. The following text has been added to the introduction to explain this relationship between the studies (Line 130-142). 

Given that this play-based PMI achieved its primary aim of improving the playfulness of children with ADHD, further exploration of the intervention is warranted to understand the effects of the intervention from a more holistic perspective. Critical to the intervention was the inclusion of a typically-developing peer in every clinic-based intervention session as well as home-based activities between clinic visits. Throughout their participation peers were encouraged to support the play of children with ADHD, and model play behaviours that lead to mutual enjoyable social play experiences. Ensuring that there are also positive effects on the playfulness of these peers, or in the least there are no detrimental effects, is critical given that they and their families are dedicating resources to the intervention.

2. Introduction: There are a number of statements within the introduction that need supporting references. page 3, line 52; page 3 line 54; page 3, line 73. page 5, line 109

Response: References are now provided for the statements referred to here.

3. Line 79 " As a reader I'm not convinced by the afore introduction that social challenges warrant the development of interventions to increase play participation. "

As a reader I'm not convinced by the afore introduction that social challenges warrant the development of interventions to increase play participation. So far the argument in the introduction has stated difficulties in social situations and friendships but there is no introduction about children with ADHD NOT participation in play - just that they have difficulties with social interactions during play, or may choose, or be directed by families, to play within their own family social situations. There is some evidence that solitary play is predictive of reduced anxiety and depression in ADHD, and also some evidence that children with adhd are more fun. Can the authors include a balanced view in their introduction, and particularly identify that it is children with ADHD AND difficulties with making and retaining friendships that was the focus of their research.

Response: We understand with and agree with the points the reviewer is making here about the play of children with ADHD, however the focus of this paper is social play so we have opted to continue to focus on play with peers rather than solitary play.

We believe the issue raised by the review related to the use of the word “participation” in the refenced sentence. We agree – participation in play is not the focus of this research. Rather, it’s the differences in social play skills in combination with the skills and perceptions of others that we are addressing in order to support the development and maintenance of friendships. As such, we have adjusted the referenced sentence to the following (Line 79-80)

The social challenges experienced by children with ADHD warrant the development of evidence-based interventions that focus on the social play skills of children with ADHD in conjunction with those of their usual playmates.

4. METHODS: Well described. Page 8, line 180-181 "Neither parents nor teachers raised concerns about the playmate participant’s development. Were parents and teachers specifically asked about behaviours of the playmate or are you saying that they just didn't rise any concern. There is a difference and this should be made clear. For example, parents and teachers may not raise concerns voluntarily because it would be inappropriate to share such information.

Response: Parents were explicitly asked as part of the intake process whether they or their child’s teacher had any concerns about their child’s social skills, behaviour, academic progress. The manuscript has been updated with the following text to make this distinction explicit (Line 195-197)

Parents reported that neither they nor their child’s teacher had concerns about the playmate participant’s social skills, behaviour or academic development

5. Instrument Test of Playfulness: Is there a value for a clinically meaningful change score since this is your main outcome of interest? Even more useful would be a value of worthwhile change. Is the change been anchored against a worthwhile change outcome.

Response: Currently there is no evidence for clinically meaningful change score values for the ToP. Instead, we have reported effect sizes as an indicator of the magnitude of change in playfulness.

In lieu of a value for worthwhile change, we have also added quotes that children used during play that demonstrated expert performance in the areas of playfulness where change was observed and measured (see Methods: Line 222-225, Results: Lines 418-426) .

6. Procedure and Intervention well described.

RESULTS: As per above: Is there a clinically meaningful change for ToP? This would be helpful in interpreting the change scores from a clinical perspective. Similarly it would be good to understand a worthwhile change? Was the ToP anchored to any assessment for this?

Response: Please see response to 5 above.

7. I am not clear on teh value of looking at the playmates change scores r correlation with their CCBRS since their CCBRS is not within the clinical range, and the correlations are not strong (not unexpected since they are not within the clinical range.

Response: We saw value in exploring the associations between the behavioural characteristics of the playmates and the benefits they received in their play skills, as while on average the CCBRS scores of the playmates were not within the clinical range, a degree of variability was still present in that non-clinical range. Understanding the behavioural profile of playmates in relation to the change in play skills begins to provide insight into the characteristics to consider when families are selecting playmates to participate in the future. It also provides some direction as to the characteristics of children who may not benefit as much from participating as a playmate and may not be the ideal playmate to invite as they may not have the same benefit for their time as other children.

8. DISCUSSION: The discussion focusses too much on the ADHD population rather than the main am of this sub study.

Response: The discussion has been revised in consideration of this comment and the comments below. We believe that through these revisions the discussion now has a stronger focus on the aims of this study.

9. Line 413 -414 is is one of the first studies to report on the intervention outcomes of the peers of children with ADHD following a PMI."

Just because something is do e first doesn't mean it is of value. It doesn't add any value to your study by stating this so I would remove it. 

Response: Per the reviewer’s suggestion, this sentence has been removed.

10. How does improvements in play of typically developing children support including them in interventions for ADHD? please elaborate on your statement around this.

Response: We argue that while PMIs are beneficial to children with ADHD, if participation in a PMI is of no benefit to the peer, then the argument for including these usual playmates in PMIs is weakened. Our findings indicate that, for this PMI at least, playmates benefit along with children with ADHD, strengthening the argument for their inclusion in PMIs. Elaboration has been provided within the manuscript (Lines 450-452)

11. Line 421-423 " As a result of these improvement, the preferred playmates of children with ADHD are likely to be better equipped to deal with the challenging behaviours often presented by children with ADHD to maintain ongoing and mutually enjoyable social interactions"

I don't agree with this statement. There is no outcome showing that these playmates become better equipped to deal with challenging behaviour since this was not measured in this study. All that you can say is that the intervention is effective in typically developing children.

Response: The reviewer is correct, we did not measure challenging behaviour per se in this study, and this statement has been adjusted to focus on play behaviours rather than challenging behaviours more broadly (Lines 455-458).

12. Line 426 -428 The benefits for the typically developing peers included reduced negative sociometric nominations from peers in the program, increased reciprocated friendships, and reduced negative interactions.

Are these the results from your study? Or Mikami. It is not clear. If from Mikami how does it relate to your outcomes since you didn't report on these?

Response: The statement referred to in this comment is in relation to Mikaimi’s study and not our current study. The sentence has been adjusted to make this distinction clearer (Line 494)

13. Line 435-483 These paragraphs should go first in your discussion since it was the main aim of this study. I suggest you move them all up.

Response: Thank you for this suggestion. Those paragraphs have been moved to the beginning of the discussion.

14. Remove the "quotations" from discussion. These are qualitative results that you have not addressed in methods or results. The last paragraph 484-490 

Response: These quotes were originally included to provide further illustration of the observed ways that playmates supported and prompted the play of children with ADHD. We understand the reviewer’s point that these qualitative examples are not addressed within the design of the study and have removed them from the discussion. However, per our response to the reviewer’s comment 5, the collection and presentation of these quotes have now been added to the methods and results as qualitative support for the quantitative data.

15. This last paragraph in the discussion is not related to playmates outcomes so I can't see the relevance of including it.

Response: We thank the review for this feedback and have removed the paragraph accordingly.

16. Conclusion: The last lines 507-511 are not a conclusion of this study.

Response: The conclusion has been revised to align more closely with the aims and findings of this study.

---

## [Decision Letter · Decision Letter 1]

7 Oct 2022

A Pairwise Randomised Controlled Trial of a Peer-Mediated Play-Based Intervention to Improve the Social Play Skills of Children with ADHD: Outcomes of the typically-developing playmates

PONE-D-22-07013R1

Dear Dr. Cordier,

We’re pleased to inform you that your manuscript has been judged scientifically suitable for publication and will be formally accepted for publication once it meets all outstanding technical requirements.

I do want to personally apologise for the delay in returning the results of the review. I had a very difficult time securing two reviewers and have made the decision to go with the one review

Kind regards,

Amanda A. Webster

Academic Editor

PLOS ONE

Additional Editor Comments (optional):

Reviewers' comments:

Reviewer's Responses to Questions

**Comments to the Author**

1. If the authors have adequately addressed your comments raised in a previous round of review and you feel that this manuscript is now acceptable for publication, you may indicate that here to bypass the “Comments to the Author” section, enter your conflict of interest statement in the “Confidential to Editor” section, and submit your "Accept" recommendation.

Reviewer #1: All comments have been addressed

2. Is the manuscript technically sound, and do the data support the conclusions?

Reviewer #1: Yes

3. Has the statistical analysis been performed appropriately and rigorously? 

Reviewer #1: Yes

4. Have the authors made all data underlying the findings in their manuscript fully available?

Reviewer #1: No

5. Is the manuscript presented in an intelligible fashion and written in standard English?

Reviewer #1: Yes

6. Review Comments to the Author

Reviewer #1: (No Response)

7. PLOS authors have the option to publish the peer review history of their article (what does this mean?). If published, this will include your full peer review and any attached files.

Reviewer #1: No

---

## [Editor Report · Acceptance letter]

12 Oct 2022

PONE-D-22-07013R1 

A pairwise randomised controlled trial of a peer-mediated play-based intervention to improve the social play skills of children with ADHD: Outcomes of the typically-developing playmates 

Dear Dr. Cordier:

I'm pleased to inform you that your manuscript has been deemed suitable for publication in PLOS ONE. Congratulations! Your manuscript is now with our production department. 

Kind regards, 

on behalf of

Dr. Amanda A. Webster 

Academic Editor

PLOS ONE